# Altered Envelope Structure and Nanomechanical Properties of a C-Terminal Protease A-Deficient *Rhizobium leguminosarum*

**DOI:** 10.3390/microorganisms8091421

**Published:** 2020-09-16

**Authors:** Dong Jun, Ubong Idem, Tanya E. S. Dahms

**Affiliations:** Department of Chemistry and Biochemistry, University of Regina, Regina, SK S4S 0A2, Canada; andrea.jun.dong@gmail.com (D.J.); ubidem@gmail.com (U.I.)

**Keywords:** atomic force microscopy, cell envelope, C-terminal protease, force spectroscopy, *Rhizobium leguminosarum*, viscoelasticity

## Abstract

(1) Background: Many factors can impact bacterial mechanical properties, which play an important role in survival and adaptation. This study characterizes the ultrastructural phenotype, elastic and viscoelastic properties of *Rhizobium leguminosarum* bv. *viciae* 3841 and the C-terminal protease A (*ctpA*) null mutant strain predicted to have a compromised cell envelope; (2) Methods: To probe the cell envelope, we used transmission electron microscopy (TEM), high performance liquid chromatography (HPLC), mass spectrometry (MS), atomic force microscopy (AFM) force spectroscopy, and time-dependent AFM creep deformation; (3) Results: TEM images show a compromised and often detached outer membrane for the *ctpA* mutant. Muropeptide characterization by HPLC and MS showed an increase in peptidoglycan dimeric peptide (GlcNAc-MurNAc-Ala-Glu-meso-DAP-Ala-meso-DAP-Glu-Ala-MurNAc-GlcNAc) for the *ctpA* mutant, indicative of increased crosslinking. The *ctpA* mutant had significantly larger spring constants than wild type under all hydrated conditions, attributable to more highly crosslinked peptidoglycan. Time-dependent AFM creep deformation for both the wild type and *ctpA* mutant was indicative of a viscoelastic cell envelope, with best fit to the four-element Burgers model and generating values for viscoelastic parameters k_1_, k_2_, η_1_, and η_2_; (4) Conclusions: The viscoelastic response of the *ctpA* mutant is consistent with both its compromised outer membrane (TEM) and fortified peptidoglycan layer (HPLC/MS).

## 1. Introduction

Mechanical changes arising from the environment have been linked to microbial gene expression, physiology and pathogenesis, and bacterial mechanical properties are integral to both their survival and adaptation [1]. The microbial envelope is viscoelastic [2], exhibiting both elastic and viscous properties, with enough flexibility to accommodate growth and division but sufficiently rigid to resist turgor pressure and maintain a specific shape [3]. While an elastic material can recover its shape following instantaneous deformation, a viscous one responds with a flow rate [4], and thus a viscoelastic material can both store and dissipate mechanical energy in response to stress [5].

The mechanical properties of the bacterial cell envelope are defined by its biochemical composition and interactions between the various components [6]. The cell envelope in Gram-negative bacteria is composed of three principal layers, the inner membrane, periplasm, and outer membrane [7,8]. The inner membrane (IM), which surrounds the bacterial cytosol, is composed of a phospholipid bilayer that houses many proteins associated with energy production, lipid biosynthesis, protein secretion, and transport [9,10]. The periplasm is more viscous than the cytoplasm [11], with densely packed soluble proteins [12] and the peptidoglycan polymer which is composed of an alternating chain of N-acetyl glucosamine (GlcNAc) and N-acetyl muramic acid (MurNAc) covalently linked by short peptide chains [13]. The inner leaflet of the asymmetric outer membrane is composed of lipid A (two glucosamine subunits covalently linked to six or seven acyl chains and phosphate groups) and the outer leaflet contains lipopolysaccharide (LPS) [14]. LPS is lipid covalently linked to a core polysaccharide, consisting of heptose, hexoses, and 3-deoxy-D-manno-octulosonic (2-keto-3-deoxyoctonic) acid (Kdo) [15], and O-antigen composed of repetitive oligosaccharide subunits extending outward from the outer membrane [16]. Crucial for osmotic adaptation [17,18] are outer wall cyclic β-glucans, composed of a D-glucose backbone with 5–40 glucopyranosyl residues linked together by β-1,2-, β-1,3-, or β-1,6-glycosidic bonds [19]. 

All layers in the cell envelope contribute to the overall mechanical response, which includes instantaneous elastic deformation, delayed elastic deformation, and viscous flow under an applied external force [6]. The periplasmic cell wall confers rigidity, maintains cell shape, and is the major contributor to elastic properties of the bacterial cell envelope [20,21], with peptidoglycan playing a significant role [22]. The viscoelastic properties of the inner membrane depend on its chemical composition, intrinsic tension as a function of differential packing, and adaptation to physical variables, such as temperature, pressure, and pH [23]. Membrane domains of fluidly ordered phospholipids and other membrane components impact bilayer viscosity [24,25,26], where the liquid phase is less stiff than the gel phase [27]. The softer outer membrane contributes less elasticity than the periplasmic peptidoglycan layer [21], but its viscoelasticity contributes to the overall mechanical response, with viscous relaxation in *Escherichia coli* attributed to reorganization of membrane components [28]. 

Atomic force microscopy (AFM) has been used to measure the mechanical properties of the bacterial cell surface under physiological conditions, revealing that the bacterial cell envelope is extensible, flexible, elastic and viscous [4,6,22,29,30]. The viscoelastic response induced by an external force on the cell surface can be tracked using force spectroscopy, but interpretation of force curves acquired from Gram-negative bacteria is nontrivial [4]. Vadillo-Rodriguez et al. developed a model combining elastic and viscous elements to describe the deformation of the bacterial cell envelope as a time-dependent creep response under constant external loads [6]. Based on one of the simplest models to predict creep response [2], it consists of an elastic spring combined in series with a parallel spring and dashpot. Their model successfully interpreted the mechanical behavior of the cell envelope of *Pseudomonas aeruginosa* PAO1 in response to an applied constant compressive force [6] and three viscoelastic parameters, k_1_, k_2_, and η_2_, were measured for Gram-negative (*E. coli*) and Gram-positive (*Bacillus subtilis*) bacteria [22]. Both studies reveal two elastic processes contributing to the mechanical response of the bacterial cell envelope: an instantaneous elastic deformation of the cell envelope (k_1_) and a delayed elastic creep deformation characterized by k_2_ and η_2_. Lu et al. further modified the model to interpret the viscoelastic behavior of the bacterial cell envelope in response to antimicrobial peptides, showing that increased membrane permeability can alter viscoelastic properties [30].

Here we determine the mechanical properties of the *Rhizobium leguminosarum* cell envelope and evaluate the influence of a C-terminal protease A (*ctpA*) mutation on its mechanics. Gram-negative free-living rhizobia are faced with a dynamic and unpredictable soil environment [31,32] prior to forming root nodules [33], requiring substantial adaptation in which the cell envelope plays a significant role. Carboxyl terminal protease (Ctp) cleaves a number of amino acid residues from the carboxyl terminal end of protein precursors during post-translational modification [34]. The first homolog of Ctp in the bacterial kingdom was discovered in *E. coli* and designated as tail-specific protease (Tsp), also known as Prc. The *prc* deletion mutant was defective in processing the penicillin-binding protein 3 (PBP3), exhibited thermosensitive growth in salt-free media, had a filamentous morphology, reduced cell wall integrity, and caused leaky membranes [34]. Disruption of *prc* increases antibiotic susceptibility [35] and reduces the level of bacteremia in human sera, increasing susceptibility for serum killing of the pathogenic *E. coli* strain RS218 [36], thus linking Ctp to the regulation of cell morphology, cell survival, and host evasion. The *R. leguminosarum ctpA* null mutant grows in complex liquid medium but not complex semi-solid agar and has increased susceptibility to detergent, indicating a compromised cell envelope [37]. We found the *ctpA* mutant incapable of developing fully mature biofilms, consistent with its altered surface ultrastructure, greater roughness, and stronger adhesion to hydrophilic surfaces [38]. We since identified nine periplasmic solute-binding components of ABC transporters with altered abundance in the *ctpA* mutant [39], consistent with *CtpA* as a periplasmic enzyme [34]. Thus, we hypothesized that a penicillin binding protein (PBP) is a *CtpA* substrate, and its absence would alter the mechanical properties of the rhizobial cell envelope by changing the peptidoglycan structure. This study not only offers insight into the *ctpA* mutation, but helps develop an understanding of a complex microbial interface and the intricate relationship between rhizobial cell envelope structure, dynamics, and function.

## 2. Materials and Methods

### 2.1. Bacterial Strains, Plasmids, and Media

*R. leguminosarum* biovar *viciae* 3841, a spontaneous streptomycin-resistant derivative of *R. leguminosarum* bv. *viciae* strain 300 [40], served as the wild-type strain used in this study. The *ctpA* mutant strain 3845 was prepared by Gilbert et al. [37] and strains of *R. leguminosarum* were cultured in Vincent’s minimal medium (VMM) [41] with 10 mM mannitol as a carbon source. Media was supplemented with appropriate antibiotics (µg/mL): streptomycin (500); neomycin (100); tetracyclineD (5). Unless otherwise stated, all materials were purchased from Sigma Aldrich (Oakville, ON, Canada).

### 2.2. Isolation of Peptidoglycan

Peptidoglycan was prepared as described by Pisabarro et al. [42]. Briefly, cell cultures (500 mL), initiated from a 10 mL starter culture, were grown (OD_600_ ~ 0.6) and quickly cooled in an ice/water bath. Cells were harvested by centrifugation (20 min, 8000× *g*, 4 °C), resuspended in 2 mL VMM buffer (5.74 mM K_2_HPO_4_, 7.35 mM KH_2_PO_4_, 5.93 mM KNO_3_, 0.04 mM FeCl_3_·6H_2_O, 1 mM MgSO_4_·7H_2_O, 0.46 mM CaCl_2_·6H_2_O) which was isotonic to the VMM broth, the suspension boiled in sodium dodecyl sulfate (SDS, 8%, 4.5 mL) for 60 min, kept at room temperature (RT) overnight and boiled again for 60 min. The suspension was ultracentrifuged (100,000× *g*, 45 min), the pellet washed with hot deionized water (5×), high molecular weight glycogen removed by treatment with 200 µg of α-amylase (1 h at 37 °C) and covalently attached proteins removed with protease (200 µg, 1 h, 60 °C). The insoluble peptidoglycan was recovered by ultracentrifugation (100,000× *g*, 30 min) and lyophilized overnight.

### 2.3. Muropeptide Structural Analysis

Muropeptides were prepared as described by Pisabarro et al. [42]. Briefly, dry peptidoglycan was diluted (2 mg/mL) in 25 mM sodium phosphate (pH 6.8), treated with lysozyme (20 µg/mL; 37 °C O/N), followed by boiling (10 min, 100 °C), and centrifugation (19,000× *g*, 10 min) to remove insoluble material. Supernatant was mixed with the same volume of sodium borate buffer (0.5 M; pH 9.0) and incubated (RT, 30 min) with excess borohydride, neutralized with 20% phosphoric acid (1/20 volume) to a pH between 4 and 5, and samples stored at −20 °C. Muropeptides were separated on a reversed-phase column (3.5 µm ZORBAX, 300SB-C18, 150 × 4.6 mm, Agilent, Santa Clara, California, United States) using an Agilent high performance liquid chromatography (HPLC) system consisting of a Hewlett Packard 1050 series pump and a UV detector. Initial separation used gradient elution from 100% buffer A (H_2_O with 0.1% TFA *v*/*v*) to 100% Buffer B (acetonitrile with 0.1% TFA *v*/*v*) over 120 min at a flow rate of 1 mL/min and conditions were optimized to separate all the rhizobial muropeptides in a single run (Appendix A). The muropeptide fractions were detected at 205 nm, collected at full width at half maximum (FWHM), and quantified by integration of their peak area using ChemStation (Agilent). The total peak area from all fractions, excluding the salt fraction, was normalized to 100%, and the percent total area of each fraction determined. 

Muropeptide fractions were analyzed on an AB 4800 matrix-assisted laser desorption/ionization time-of-flight (MALDI-TOF) mass spectrometer (Applied Biosystems, LLC, Frederick, MD, USA) [43]. For MALDI matrix deposition, 10 mg/mL 2,5-dihydroxybenzoic acid was prepared in methanol/water (1:1 *v*/*v*) with 50 mM of NaCl, to ensure that sodiated adducts [M + Na]^+^ were the predominant ions. Matrix solution (0.8 μL) was deposited onto a 384-well plate (Opti-TOF insert, Applied Biosystems), followed by the application of the muropeptide sample (1.0 μL in 5% acetonitrile). The plate was dried under ambient conditions, loaded into the instrument operated with the Applied Biosystems 4000 Series Explorer program (Version 3.5.3) and data collected in the positive ion reflection mode. A six-peptide reference mixture was used as an external mass calibration standard prior to data acquisition. The ion extraction delay time was set to be 500 ns, and 1000 laser shots were collected and averaged for each spectrum. 

### 2.4. Transmission Electron Microscopy (TEM) Imaging 

Single colonies of *Rhizobium leguminosarum* wild type 3841 and *ctpA* mutant 3845 from a TY agar plate, supplemented with 20 mM mannitol were isolated and grown in 5 mL TY broth medium at 30 °C to an OD_600_ of 0.6, centrifuged (1000× *g* for 5 min) and cell pellets resuspended in VMM buffer. Cells were fixed (3.7% formaldehyde and 0.2% Triton X-100 in VMM buffer; 20 min, 30 °C), rinsed with VMM buffer, and transported to the transmission electron microscopy (TEM) imaging facility (University of Calgary) where samples were fixed (1 h) with 1% osmium tetroxide, rinsed, substituted with 50%, 75%, 95%, and 100% acetone, embedded with JEMBED liquid plastic (Canemco, Lakefield, Québec, Canada), and small blocks of embedded bacteria cut with an ultramicrotome (Leica ultracut). Ultrathin sections were then positively stained with uranyl acetate and lead citrate for TEM (H-7650, Hitachi, Tokyo, Japan) imaging at 60 keV. 

### 2.5. AFM Imaging and Force Data Acquisition 

Cell pellets, as prepared for TEM, were resuspended in buffer prior to immobilization. Topographic features of pea leaves, initially tested as natural substrates for live *R. leguminosarum*, were not discernable from rhizobia in AFM images, so live, actively growing samples, were prepared on dialysis tubing using the method adapted from our previous work on *A. nidulans* [44]. Briefly, sterilized dialysis tubing was incubated for 24 h in *R. leguminosarum* culture and then rinsed with VMM buffer to remove non-adhering rhizobial cells. During imaging, media was fed to the rhizobia through the filter paper inserted in the dialysis tubing to maintain cell viability. Images are representative of rhizobial cells from three biological replicates, at least 10 images per replicate and a minimum of three cells per image. 

Preliminary experiments probing the mechanical properties of *R. leguminosarum* wild type 3841 and *ctpA* mutant 3845 used a commercial AFM (TopoMetrix Explorer 2000 AFM, Veeco Instruments, Santa Barbara, CA) equipped with a dry AFM scanner (Model No.400006). Force curves were acquired using V-shaped silicon nitride tips (Model MLCT-EXMT-A1, Veeco) having a nominal spring constant of 0.05 N/m. Force-indentation experiments were carried out using AFM force spectroscopy at room temperature. The spring constant of the cantilever was determined directly prior to force curve acquisition according to Cleveland and coworkers [45]. The AFM tip approached the sample at 10 µm/s until it reached a preset position and then retracted. The slopes of the force curves were recorded and converted into the bacterial spring constant according to the equation:(1)$$k_{b}=\frac{{\mathrm{sk}}_{c}}{1-s};$$
where s is the slope of the force curve and k_c_ is the spring constant of the cantilever. Surface adhesion values were measured from the point at which the AFM tip retracts from the sample surface to zero adhesive force. The force curves acquired from the experiments were processed with SPMLab software 5.01.

### 2.6. AFM Creep Experiments 

Bacterial cells resuspended in media (100 µL) were deposited onto PLL coated glass coverslips [38], and incubated (30 min, RT). The bacterium-coated glass coverslip was gently rinsed with nanopure water five times to remove excess media and loosely attached bacterial cells in preparation for AFM imaging and force spectroscopy. To rule out a deleterious effect of PLL on rhizobial viability, growth medium was added to the sample and bacterial viability monitored over 24 h, at RT in a separate experiment, during which we observed rhizobia dividing, detaching, and sometimes reattaching to the surface [6].

To determine the viscoelastic responses of *R. leguminosarum* wild type 3841 and *ctpA* mutant 3845, AFM (Nanowizard 3, JPK, Berlin, Germany) measurements were carried out at room temperature under nanopure water (resistivity > 18.2 MΩ) to maximize osmotic pressure (Rodriguez et al., 2008). Live rhizobial cells were first imaged in quantitative imaging mode at low applied force (<0.5 nN) at a scan rate of 0.5 Hz using silicon nitride V-shaped cantilevers having a pyramid-shaped tip with a typical radius of curvature of 10 nm (PNP-TR, nominal spring constant = 0.08 N/m, Nanoworld, Neuchâtel, Switzerland). Prior to force experiments, the cantilever spring constant was determined using the thermal fluctuation method [46]. To determine the elastic and viscous contributions to the mechanical properties, three force curves were collected from the top center of 5–10 single cells from at least three biological replicates, with a constant tip approach rate (1 µm/s) set to a predetermined loading force value (2, 4, 6, and 10 nN). Once the loading force was reached, it was held constant for 10 s during creep experiments. The fast elastic response and delayed elastic response of the cell envelope, corresponding to creep deformation, were described by the four element Burgers model [30]:(2)$$Z\left(t\right)=\frac{F_{0}}{k_{1}}+\frac{F_{0}}{k_{2}}\left(1-e^{-\frac{k_{2}}{\eta_{2}}t}\right)+\frac{F_{0}}{\eta_{1}}t;$$
where Z(*t*) is the total deformation of the material at time *t*, *F*_0_ is the applied force, *η*_1_ and *η*_2_ are dashpot viscosities for which *k*_1_ and *k*_2_ are parallel spring constants. When a constant force is applied to the system at time *t*, the deformation can be described as: (3)$$Z\left(t\right)=Z_{s}+Z_{d}=\frac{F_{0}}{k_{1}}+\frac{F_{0}}{\eta_{1}}t;$$
where *Z_s_* and *Z_d_* are the deformation of the spring and dashpot, respectively. The best-fit values of *k*_1_, *k*_2_, *η*_1_, and *η*_2_, describing the mechanical response of the cell, were obtained by nonlinear regression in Excel (Microsoft office 2010). Differences in parameters between the wild type and *ctpA* mutant *R. leguminosarum* were assessed with a paired Student’s *t*-test.

## 3. Results

### 3.1. Cell Envelope Ultrastructure 

Prior evidence of cell envelope disruption for the *ctpA* mutant came from AFM images of fixed cells [38]. Since AFM-based force spectroscopy and creep deformation experiments necessitate live cells, it was important to first test if live *ctpA* mutant cells (Figure 1C,D) had less well ordered surfaces than wild type (Figure 1A,B), like their fixed counterparts. This was the case, as evidenced by the difference in topographic height range (z, color scale B,D) and subunit size (x,y). The nonlinear response often observed for Gram-negative bacteria [6] was rarely observed in the force curves of wild type cells in this study (Figure 1E). To further pinpoint specific regions of the cell envelope associated with the cell surface architectural changes, transmission electron microscopy (TEM) was used to image *R. leguminosarum*. Consistently, TEM images showed the *ctpA* mutant to have a significantly greater number of cells (low resolution images on left) with detached outer membranes (red arrows on low and high resolution images), compared to wild type (Figure 2).

### 3.2. Muropeptide Composition 

To provide preliminary evidence for the hypothesis that PBPs are target substrates of *ctpA* in *Rhizobium leguminosarum* bv. *viciae* 3841, muropeptide was analyzed. Muropeptides isolated from lysozyme-treated peptidoglycan and purified from the *R. leguminosarum* wild type and *ctpA* mutant strains had identical HPLC profiles (Appendix A), showing that the basic peptidoglycan unit was not significantly altered in the mutant. Interestingly, peaks 2, 10, and 11 from the *ctpA* mutant strain had significantly larger peak area (*p* < 0.05, *n* = 10) than wild type (Table 1). 

MALDI-TOF mass spectrometry (MS) analysis of the major components (peaks 2, 3, 6, 10, 11, and 12) showed identical mass spectra for corresponding peaks from wild type and *ctpA* (Appendix A, Table 1), confirming identical muropeptides eluting at the same retention time. Molecular weights (Table 1) indicate a glycan strand with repeating GlcNAc and MurNAc disaccharides, the latter linked to a 3–5 residue peptide (Appendix A). Under our experimental conditions, GluNAc-MurNAc-Ala-Glu-mDAP-Ala was the major peptide bridge component for *R. leguminosarum* bv. *viciae* 3841. The *ctpA* mutation affected the relative amounts of peptidoglycan cross linkers, namely the monomeric and dimeric peptides (GlcNAc-MurNAc-Ala-Glu-meso-DAP, GlcNAc-MurNAc-Ala-Glu-meso-DAP-Ala, and GlcNAc-MurNAc-Ala-Glu-meso-DAP-Ala-meso-DAP-Glu-Ala-MurNAc-GlcNAc), but their structural components were not altered. 

### 3.3. R. leguminosarum Cell Envelope Elasticity and Spring Constant 

To test the hypothesis that the deletion of *ctpA* has an impact on the cell envelope mechanical properties of *R. leguminosarum*, cells were first indented with AFM tips to generate force versus distance curves (Figure 1E). The cell spring constant can be calculated directly from the slope (force/distance) of force curves in the linear approach region (nN/nm). For actively growing cells in media, the *ctpA* mutant had a spring constant (0.0054 ± 0.0003 N/m (nN/nm)), k_b_, two-fold that of wild type (0.0024 ± 0.0002 N/m), indicating that the mutant has a more rigid cell envelope. Spring constants of live cells are lower by an order of magnitude compared to that of fixed samples, indicating a more elastic envelope for fixed cells, as expected [5,6]. The Hertz model, frequently used to fit force approach curves to elastic modulus of the bacterial cell envelope, is best suited to an ideal shell rather than a bacterial cell envelope, which is both anisotropic and heterogeneous, so it was not used to fit this data.

Elastic stiffness (force-displacement) can also be calculated from a nanoscale creep deformation experiment for which the initial force is rapidly applied to the microbe. To understand the response of different cell envelope components to mechanical indentation of the cell surface by the AFM tip, a range of forces were used to generate a series of force approach curves which can be used to plot loading force versus indentation, for which slope corresponds to sample stiffness. For a stiff sample like glass, with no measurable indentation, force versus indentation appears as a vertical line (Appendix A). On the other hand, the *ctpA* mutant shows a nonlinear increase in indentation up to a loading force of 4 nN, after which it was linear, and appears stiffer than wild type cells, which had an approximately linear response over the entire range of loading force (Appendix A). Interestingly, Appendix A shows that the *ctpA* mutant, compared to wild type, can be indented more easily at smaller loading forces, but less easily once the AFM tip has moved further into the cell at larger loading forces, consistent with a disrupted outer membrane (Figure 2) and a fortified peptidoglycan layer (Table 1). The spring constant, k_1_, of the bacterial cells can also be directly calculated from the ratio between the loading force and indentation depth during the linear response of a cell to loading force (Figure 3). At 2 nN, the effective cell spring constants were 0.19 ± 0.12 N/m and 0.25 ± 0.17 N/m for the wild type and *ctpA* mutant cells, respectively, statistically identical to the values at 4 nN, but they gradually increased to 0.27 ± 0.14 N/m and 0.37 ± 0.21 N/m, respectively, at 10 nN. The *ctpA* mutant has a significantly (*p* < 0.05, *n* = 100) larger spring constant than wild type at all loading force values, indicating it is stiffer than wild type, consistent with the force curve data. The average value of k_1_ obtained from the creep deformation experiments, however, is 100 fold greater than the cell spring constant measured from force-indentation curves, attributable to the different experimental conditions used to measure *k_b_* and k_1_: cells on dialysis tubing (soft) in media (hypertonic) with a fast approach (10 µm/s) on the Explorer AFM, versus those on glass (hard) in pure water (hypotonic) with a slower approach (1 µm/s) on the JPK AFM, respectively. Nonetheless, both k_b_ and k_1_ show that the ctpA mutant cells are stiffer than wildtype.

### 3.4. Viscoelasticity of the Cell Envelope 

The impact of CtpA on cell envelope viscous and elastic elements was determined from the nanoscale creep deformation experiment (Appendix A) developed by Lu et al. as a reproducible way of measuring bacterial cell viscoelastic properties [30]. Once a loading force has been rapidly applied to the cell surface (Section 3.3), it is held constant for a given period [47], revealing a time-dependent deformation indicative of a viscoelastic cell envelope (Appendix A). Since deformation of both strains under a constant loading force for 10 s proportionally increased as a function of loading force, creep deformations were conducted in the primary and secondary viscoelastic regime to probe linear and nonlinear responses [30]. The four-element Burgers model [47] was used to treat the creep deformation data of both the wild type and *ctpA* mutant, generating best-fit values of viscoelastic parameters k_1_, k_2_, η_1_, and η_2_ (Table 2).

To evaluate the effect of the *ctpA* mutation on mechanical response, histograms of k_1_, k_2_, η_1_, and η_2_ (Figure 4) were generated at low loading forces (2 and 4 nN). At 2 nN, k_1_ for the *ctpA* mutant is not significantly different from wild type, but interestingly its distribution shifts to a larger value at 4 nN and remains constant from 6 to 10 nN (Appendix A). The k_2_ and η_2_ histogram distributions are narrow at 2 nN for both the wild type and *ctpA* mutant (Figure 4), with η_2_ starting close to zero, and both broadening and shifting to higher values as a function of loading force (Appendix A).

## 4. Discussion

*CtpA*, a periplasmic protease, is critical for the maturation of proteins involved in cellular physiology, including cell envelope structure and function [37,38,39]. In the absence of *ctpA*, *R. leguminosarum* has a significantly greater number of detached outer membranes (Figure 2) and consistently less well ordered surfaces than wild type (Figure 1), consistent with a disrupted cell envelope [36,38]. The bacterial cell envelope plays an important role in maintaining cell structure [48,49], as does lipoprotein mediated linkages between the outer membrane and the peptidoglycan layer in Gram-negative cells [50]. Hara et al. attributed the deleterious effects of the Prc (*E. coli CtpA homolog*) mutation to the potential role of Prc in the maturation of PBP3, responsible for proper peptidoglycan crosslinking [34]. As such we hypothesized that *R. leguminosarum* PBP could be a substrate of *ctpA* and examined the muropeptide component of the cell wall. The higher proportion of the dimeric peptide (GlcNAc-MurNAc-Ala-Glu-meso-DAP-Ala-meso-DAP-Glu-Ala-MurNAc-GlcNAc) in the *ctpA* mutant peptidoglycan, indicative of increased crosslinking between peptidoglycan strands, may indeed compensate for the loss of outer membrane integrity associated with the *ctpA* gene deletion. However, with no change in the structure of the peptidoglycan peptide bridge for the *ctpA* mutant, this means that PBPB and PBPC are still functional in the absence of *ctpA* in *Rhizobium leguminosarum* bv. *viciae* 3841, meaning that either these PBPs are not substrates of *ctpA* or there is functional redundancy in the *R. leguminosarum* genome, with more than one C-terminal protease.

KEGG analysis [51] indicates three possible peptidoglycan biosynthetic pathways for *Rhizobium leguminosarum* bv. *viciae* 3841, each of which would result in different peptidoglycan peptide bridges. Several enzymes participate in crosslinking GluNAc-MurNAc-Ala-Glu-mDAP-Ala to other muropeptides, including MtgA, PbpC, PbpF, PbpB, and Ala-Ala-carboxylpeptidase. PbpB and PbpC could play major roles in muropeptide cross linking in *Rhizobium leguminosarum* bv. *viciae* 3841. Keiler’s study [52] of *E. coli* Tail-specific protease (Tsp (Prc); homolog of *ctpA* in rhizobia) indicates the greatest preference for Alanine (A), Valine (V), and Serine (S) at the P1 and P1′ positions, with second preference to Isoleucine (I), Leucine (L), Arginine (R), and Lysine (K). Thus, it is plausible that PbpC and PbpB from *R. leguminosarum* bv. *viciae* 3841 are cleaved by *ctpA* (Appendix A), but if they are still processed in the absence of *ctpA*, there must be another Ctp at work. A protein BLAST [53] of the *R. leguminosarum* genome identified a gene located at the 285,840–305,839 bp position in the pRL12 plasmid encoding a protein with 43% sequence identity to Tsp (Prc) of *E. coli*, a potential candidate for Ctp cleavage of PbpC and PbpB, and functional Ctp redundancy. 

The higher degree of peptide crosslinking in the mutant reinforces its cell wall, likely to compensate for the disrupted outer membrane, as an adaptive response. These major structural differences were expected to have significant mechanical consequences to the local viscoelastic properties of the *R. leguminosarum* cell envelope, which were examined by monitoring its time-dependent motion in response to a constant compressive force [6]. The nonlinear response often associated with the force curves of Gram-negative bacteria [6] was rarely observed in the wild type *R. leguminosarum*, implying that the outer membrane and extracellular polysaccharide (EPS) play a minor role in the viscoelastic response of *R. leguminosarum* to an external force. On the contrary, a significant number of force curves from the *ctpA* mutant exhibited a nonlinear regime. *Shewanella putrefaciens*, which has a 20–130 nm capsule on its cell surface, shows a pH-dependent nonlinear deformation by force spectroscopy [54], but our *ctpA* mutant produced no additional EPS [38]. The nonlinear regions of force curves from the *ctpA* mutant were up to 30–40 nm, over which cell deformation would be dominated by the outer layer, most easily attributed to the compromised outer membrane (Figure 2) and further supported by our proteomic data which indicates disruption of outer membrane ABC transporters [39]. 

On the other hand, elasticity of the bacterial cell envelope is thought to originate from the peptidoglycan network [20]. Isolated sacculi (peptidoglycan layer) from *E. coli* are flexible and elastic, expanding and contracting without rupture [55], and readily capable of recovering their original shape upon the removal of a loading force [20]. Since glycan strands are very rigid, the elastic properties of the peptidoglycan layer largely originate from the flexibility of the crosslinking muropeptide [56,57]. Thus, the increased spring constant of *ctpA* mutant cells can be reasonably attributed to their higher degree of peptidoglycan crosslinking. Since the outer membrane is compromised in the mutant, a 4 nN deformation appears to reach the peptidoglycan layer underlying the outer membrane, where the mechanical response will be dominated by components of its strand network. The greater number of peptidoglycan subunits and peptide linkages in the *ctpA* mutant would be more capable of resisting cell indentation, as reflected by stiffer cells compared to wild type. 

The greater η_1_ value for wild type compared to that of the *ctpA* mutant at the higher loading forces (Appendix A) potentially suggests that the *ctpA* mutant has a less viscous cytosol or cytoskeletal network. The delayed elastic response can be attributed to the more liquid-like components in the cell envelope. For example, membranous phospholipid and LPS at the exterior of the bacterial cell envelope are constantly in motion, with mechanical responses comparable to a liquid. Mutants lacking O-antigen side chains of LPS in *E. coli* have smaller η_2_ values but k_2_ values identical to wild type, and in *P. aeruginosa* smaller η_2_ and k_2_ values [30] were rationalized in terms of additional periplasmic water from compromised resistance to hypotonic environments. The k_2_ and η_2_ values of an *E. coli lpp* mutant, lacking cell wall associated lipoproteins, were lower than wild type *E. coli*, confirming the contribution of lipoproteins to the delayed viscoelastic response [22]. Upon exposure to cationic antimicrobial peptides, *P. aeruginosa* cells released periplasmic proteins accompanied by the entry of water into the periplasm, evidenced by a decrease in both k_2_ and η_2_ values [30]. In this study, the increase in k_2_ and η_2_ values at higher loading forces (6–10 nN) for both wild type and *ctpA* mutant cells may indicate the AFM tip beginning to penetrate the cytoskeleton [58] or stiffness contributions from the glass substrate [59]. 

## 5. Conclusions

This study shows that the absence of C-terminal protease A in *Rhizobium leguminosarum* gives rise to altered cell envelope properties (AFM), corresponding to a disrupted outer membrane (TEM) and increased crosslinking of the peptidoglycan layer (HPLC-MALDI-TOF). Both the wild type and *ctpA* mutant exhibit a viscoelastic response to a constant external stress, which during instantaneous elastic deformation are dominated by the outer cell layer at the lowest force load and by the peptidoglycan layer at higher force loads. Thus, the viscoelasticity of the *ctpA* mutant reflects both its compromised outer membrane and fortified peptidoglycan layer. Nanoscale creep deformation data provide novel information on the mechanical properties of *R. leguminosarum*, with the four viscoelastic parameters calculated according to the Burgers model attributable to different components of the bacterial cell envelope. 

## Figures and Tables

**Figure 1 microorganisms-08-01421-f001:**
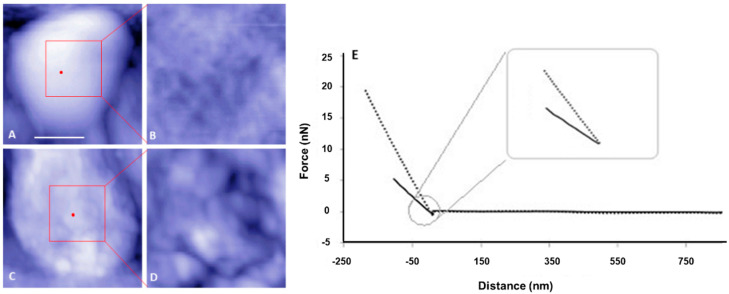
Representative atomic force microscopy (AFM) topography images and corresponding force curves of live *Rhizobium leguminosarum* bv. *viciae* 3841 (wild type, top row) and 3845 (*ctpA* mutant, bottom row). Shown are low (**A**,**C**) and high (**B**,**D**) resolution AFM images of the live wild type (**A**,**B**) and *ctpA* mutant (**C**,**D**) rhizobia. Bar for A and C is 500 nm. Red dots on A, C indicate the approximate locations at the top center of the live cells, for collecting corresponding force approach curves (**E**) with solid (wild type) and dashed (*ctpA* mutant) lines, and subsequent creep experiments.

**Figure 2 microorganisms-08-01421-f002:**
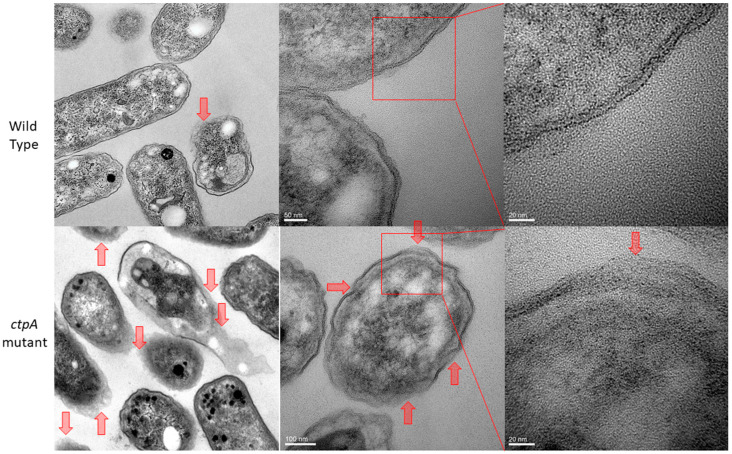
TEM images of *Rhizobium leguminosarum* bv. *viciae* 3841 (wild type, top row), the majority of which show clearly defined inner and outer membranes, and 3845 (*ctpA* mutant, bottom row). Red arrows indicate detached outer membranes, which are more numerous in the mutant.

**Figure 3 microorganisms-08-01421-f003:**
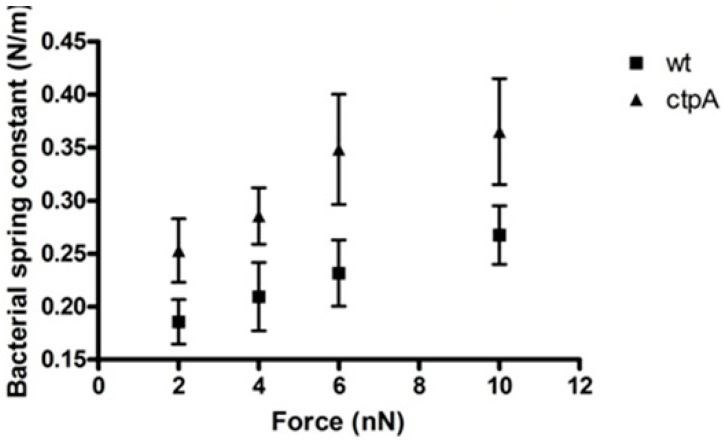
Bacterial spring constant of *Rhizobium leguminsarum* bv. *viciae* 3841 (wt) and 3845 (ctpA mutant) as a function of applied force. Spring constants were calculated from the ratio between the loading force and indentation depth during the linear response of a cell to the loading force. Values at 2 and 10 nN are statistically different (*p* < 0.05).

**Figure 4 microorganisms-08-01421-f004:**
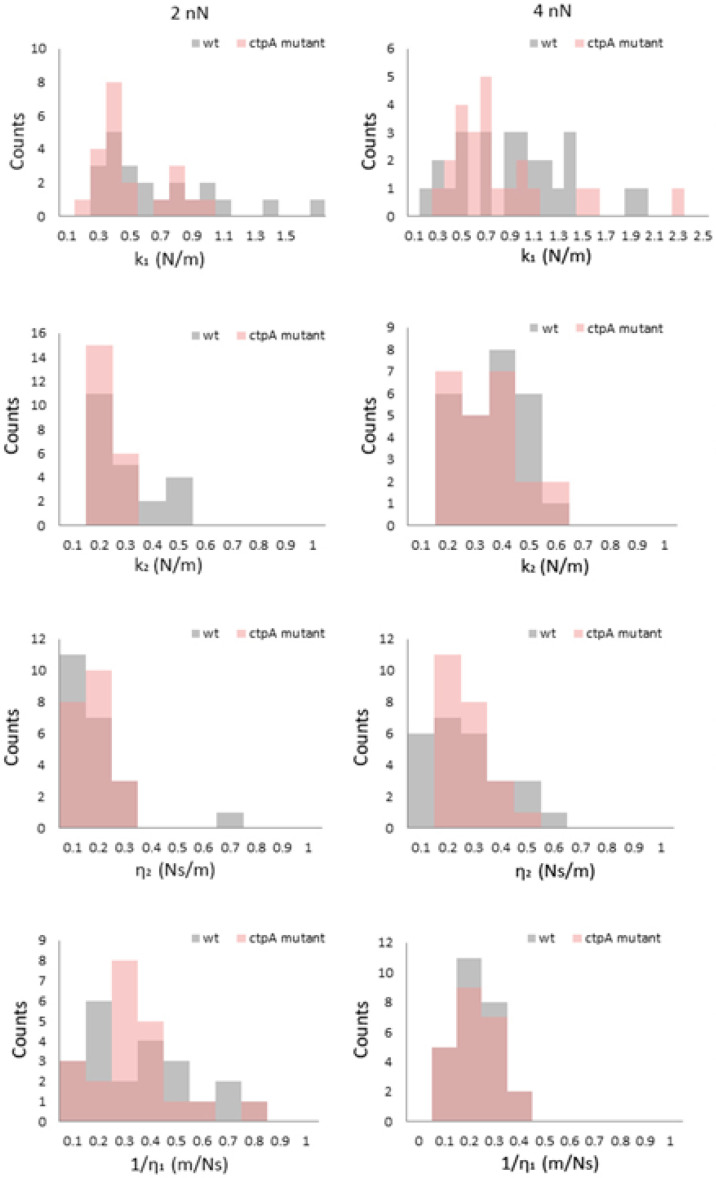
Histograms of viscoelastic parameters k_1_, k_2_, η_2_, 1/η_1_ for *Rhizobium leguminsarum* bv. *viciae* 3841 (wt, grey) and 3845 (*ctpA* mutant, pink) at low loading forces (2, 4 nN) during creep deformation experiments. k_1_ and k_2_ are elastic spring constants, and η_1_ and η_2_ are viscosity parameters from the Burgers model, in which the Maxwell and Kelvin–Voigt models are connected in series.

**Table 1 microorganisms-08-01421-t001:** Structures, molecular masses, and quantities of muropeptides from *Rhizobium leguminosarum* bv. *viciae* 3841 and *ctpA* mutant 3845.

Peak	Proposed Structure	Observed *m/z*	Calculated [M+Na]^+^	% of All Peaks	
				Wild Type	*ctp*A Mutant
2	GlcNAc-MurNAc-Ala-Glu-meso-DAP	893.4	893.8	2.0	2.7
3	GlcNAc-MurNAc-Ala-Glu-meso-DAP-Gly	950.4	950.9	1.6	1.6
6	GlcNAc-MurNAc-Ala-Glu-meso-DAP-Ala	964.5	964.9	6.2	8.3
10	GlcNAc-MurNAc-Ala-Glu-meso-DAP-Ala-meso-DAP-Glu-Ala-MurNAc-GlcNAc	1816.9	1816.8	1.3	2.4
11	MurNAc-Ala-Glu-meso-DAP-Ala-meso-DAP-Glu-Ala-MurNAc	1408.7	1408.4	1.1	2.2
12	GlcNAc-MurNAc-Ala-Glu-meso-DAP-Ala-Ala-meso-DAP-Glu-Ala-MurNAc-GlcNAc	1888.0	1887.9	2.9	3.0

**Table 2 microorganisms-08-01421-t002:** Best-fit viscoelastic constants of *R. leguminsarum* bv. *viciae* 3841 (wt) and 3845 (*ctpA*) as a function of loading force. k_1_, k_2_, η_2_, and η_1_ were fit from the four-element Burgers model [47].

Force (nN)	k_1_ (N/m)		k_2_ (N/m)		η_2_ (N•s/m)		η_1_ (N•s/m)	
	wt	*ctpA*	wt	*ctpA*	wt	*ctpA*	wt	*ctpA*
2	0.52 ± 0.37 *n* = 22	0.45 ± 0.22 *n* = 21	0.22 ± 0.12 *n* = 22	0.18 ± 0.06 *n* = 21	0.14 ± 0.12 *n* = 22	0.13 ± 0.06 *n* = 21	5.27 ± 4.38 *n* = 22	5.04 ± 4.24 *n* = 21
4	0.89 ± 0.48 *n* = 26	0.758 ± 0.46 *n* = 23	0.31 ± 0.13 *n* = 26	0.30 ± 0.13 *n* = 23	0.23 ± 0.15 *n* = 26	0.23 ± 0.10 *n* = 23	8.22 ± 6.26 *n* = 26	7.51 ± 5.58 *n* = 23
6	0.97 ± 0.56 * *n* = 18	1.39 ± 0.78 * *n* = 23	0.37 ± 0.12 *n* = 18	0.41 ± 0.16 *n* = 23	0.27 ± 0.16 *n* = 18	0.33 ± 0.23 *n* = 23	9.95 ± 5.66 * *n* = 18	7.04 ± 4.50 * *n* = 23
10	1.56 ± 0.59 *n* = 21	1.71 ± 0.92 *n* = 27	0.57 ± 0.15 * *n* = 21	0.46 ± 0.21 * *n* = 27	0.52 ± 0.36 *n* = 21	0.48 ± 0.21 *n* = 27	12.5 ± 5.97 ^†^ *n* = 21	6.78 ± 3.87 ^†^ *n* = 27

* Student’s *t*-test *p* < 0.05 (one tail); ^†^ Student’s *t*-test *p* < 0.05 (two tail).

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
