# Peer review of "Altered Envelope Structure and Nanomechanical Properties of a C-Terminal Protease A-Deficient *Rhizobium leguminosarum"

_microorganisms, 2020, doi:10.3390/microorganisms8091421_

Round 1
Reviewer 1 Report
Comments are provided according to line numbers in the submitted manuscript:
100. It is not clear how altered periplasmic binding proteins would correlate with a compromised cell envelope. These periplasmic binding proteins are presumably involved in active uptake of substrates from periplasm to cytoplasm, and these substrates may include phosphate, metal cations, peptides, amino acids, nucleotides, etc. How do these uptake functions relate to cell envelope?
105. In the introduction it is not explicitly stated that the authors hypothesize that Rhizobium CtpA modulates cell envelope properties by post-translationally modifying enzymes that control peptidoglycan structure. Is this the overarching hypothesis in the paper? Is this what is being tested? If so, it should be clearly stated in the Introduction section, and wherever appropriate thereafter.
217. The Results section does not seem to lead with questions, but rather with methods. It would be preferable to initiate each sub-section with the question being addressed or the hypothesis being tested. This suggestion arises because otherwise the data given in Figures 1 and 2 seem to lack context, and the reader has a difficult time grasping what is being tested.
220. It is recommended that the narrative be written with a less biased tone: the phrase "it was important to confirm..." might be rephrased as "it was important to test whether..."
221. Prior to citing Fig1A-D, please introduce the reader to the experiment reported in that figure. How does this experiment address a particular question?
228. (C,D) not (B,D)
235. Define PBP--this is presumed to be "penicillin binding protein," but there is an earlier reference in the Intro section to periplasmic binding proteins, which could be similarly abbreviated. Muropeptide analysis does not "determine if PBPs are target substrates of CtpA..." It does determine whether muropeptide species composition differs between wild-type and mutant cells. It is reasonable to postulate that this difference may be a consequence of CtpA acting on PBPs, but it would only be circumstantial evidence; The best evidence would come from directly assessing the C-terminal structures of PBPs in the presence and absence of CtpA.
238. In Fig. S1, please label the peak numbers, so we know what 2, 3, 6, 10, 11, and 12 correspond to.
256. Again, this is a sub-section that is in need of proper contextualization, by stating the question being investigated, and preferably clarifying how results in the previous sub-section give rise to this subsequent question. The reader will be asking: "how does one thing lead to the next thing in this paper? What is the logical flow that takes us from one experiment to the next?"
258. The in-text reference to Figure 2E seems to be in error. This should presumably refer to Figure 3, but this figure doesn't contain a Part E.
271. How does the previous paragraph (starting line 262) lead to this one? What is the logic that takes us from force v. indentation analysis to spring constants? How does one analysis build from the previous?
312. Please mention way back in the introduction that CtpA is a periplasmic protease. One presumes from the evidence that it is, but this should not be stated for the first time in the Discussion section.
320-324. Unpack this a little more. What can explain the increased dimeric peptide in the ctpA mutant? Could this be brought about by increased function of a PBP? Decreased function of some degradation enzyme?
332. In Figure S3, we see a 29-aa sequence for PBPB, and a 16-aa sequence for PBPC. Are these the C-terminal ends of these polypeptides? If so, please indicate this more clearly.
333. Figure S3 is brought up out of order. Figures S4-S7 are referred to previous to Figure S3.
338. The authors state here that CtpA probably influences peptidoglycan properties not by directly modifying PBPs, but by causing outer membrane integrity issues, and the peptidoglycan modifications come about as an indirect result, is this correct?
380. By reading this Conclusions section, a reader may be confused about whether the aim of the work was to learn more about the functional role of ctpA, or if the aim was to simply characterize certain biophysical attributes of Rhizobium, with analysis of the ctpA mutant being of minor importance.
Reviewer 2 Report
I read the manuscript with interest, because it describes the determination of parameters in bacteria using analytical methods (AFM), which I rarely found until now in microbiology. This brings me directly to my general comments: I am not an expert in Rhizobium leguminosarum and also not in elastic properties of bacterial cell walls. Thus eventually I do not properly judge the value of the data. But I guess, that I am not the only one among the readers of the manuscript and thus, I suggest that the authors modify the introduction to better explain, why these studies are important, not only in general for bacteria, but particularly for R. leguminosarum, and why they expected effects, when they used the ctpA-mutant.
More specific comments:
Fig. 1: this figure does not clearly show any significant effect. It does not become clear, what is shown on the pictures and which differences should be seen. How representative are the chosen parts? What is shown in Fig. 1E? Is Fig. 1E referred to in line 258 (now Fig. 2E, which does not exist)? In the legend of Fig. 1 the part B is mentioned twice, I guess, the second "B" should be "C"?
Fig. 2: A better description of what can be seen in the different parts of the figure, particularly of the "overview" in the first part would be helpful. The claimed "increased detached outer membrane" is not convincingly shown, because only small parts of a single bacterial cell wall are highlighted in the third parts of the figures.
Table 1: This table is used to illustrate the increased cross-linking of the peptidoglycan layer, as far as I understood. For me this is not convincing, because the amounts of the single constituents are given in "% of all peaks" without explaining whether "all peaks" are identical between both strains, or whether the number of peaks is used or the area under the peaks. Moreover the muropeptides with 2 mur-residues, i.e. those showing a cross-link by a peptide, are not so different between the wild-type and the mutant. Eventually the authors can better explain, how they analyse these data.
Chap 3.3: I am confused by the description of the spring constants: In the first and in the last paragraph of this chapter spring constants are mentioned, which clearly differ, and those from the first paragraph are not found in any figure (there is no Fig. 2E, and Fig. 1E shows nN (?), but not N/m). Finally the authors mention that different experimental set-ups may be the reason. It would be helpful, if the conditions under which the data are generated are clearly mentioned together with the data and if the need for different approaches is better explained.
In my opinion it is a significant drawback that these constants and coefficients are not validated by other methods.
2nd paragraph of the discussion: I suggest to add more information to better explain the conclusions: For me the relationship between Ctp and the list of crosslinking enzymes is not intuitively clear. From which data do the authors conclude the specific relevance of PbpC and PbpB and that PbpC is still processed in the absence of CtpA? And how does this correlate to the increased stiffness?
Round 2
Reviewer 1 Report
Previous reviewer concerns have been effectively addressed by the authors.